materials science

polymer-derived ceramic, fibres, ceramic matrix composite, ceramics, polymers

**Author for correspondence:**
Gurpreet Singh
e-mail: gurpreet@ksu.edu

# Preparation and structure of SiOCN fibres derived from cyclic silazane/poly-acrylic acid hybrid precursor

Zhongkan Ren[1], Christel Gervais[2] and Gurpreet Singh[1]

[1]Department of Mechanical and Nuclear Engineering, Kansas State University, Manhattan, KS 66506, USA
[2]Sorbonne Université, CNRS UMR 7574, Collège de France, Laboratoire de Chimie de la Matière Condensée de Paris, 75005 Paris, France

  ZR, 0000-0003-4280-2899

Ceramic matrix composite (CMC) materials have been considered a desired solution for lightweight and high-temperature applications. Simultaneously, among all different CMC reinforcements, polymer-derived ceramic (PDC) fibres have gained attention for the intrinsic thermal stability and mechanical strength with simple and cost-effective synthesis techniques. Here, carbon-rich SiOCN fibres were synthesized via hand-drawing and polymer pyrolysis of a hybrid precursor of 1,3,5,7-tetramethyl-1,3,5,7-tetravinyl-cyclotetrasilazane (TTCSZ) and poly-acrylic acid (PAA). The type of silazane reported in this work is considered as a major precursor for SiCN; however, it is unspinnable, due to its unfavourable physical properties (low viscosity) and chemical structure (cyclic rather than linear structure). The introduction of PAA to TTCSZ to create a hybrid precursor remarkably improved the spinnability of the silazane and should be widely applicable to other unspinnable PDC pre-ceramic polymers. Investigations on the structural and compositional development of the fibres were mainly conducted via Raman spectroscopy, Fourier-transform infrared spectroscopy, scanning electron microscopy, X-ray photoelectron spectroscopy, nuclear magnetic resonance and thermo-gravimetric analysis to determine spinnability, free carbon content, cross-linking and pyrolysis behaviour of the fibres, respectively.

## 1. Introduction

Over the past few decades, there has been a rising desire for new structural aerospace materials with lightweight, high modulus and high thermal stability that has motivated the advance of

materials science. Non-oxide ceramic matrix composites (CMCs) are promising because of the low density, remarkable mechanical strength and chemical resistance as well as elevated fracture toughness by fibre reinforcement. The microporous and micro-cracked matrix allows several times higher strain-to-failure than monolithic ceramics that makes it one of the optimum structural materials at high temperature [1], while reinforcement contribution is mainly towards properties such as electrical conductivity, hardness, thermal expansion and shock resistance.

Polymer-derived ceramic fibres (PDCFs) are considered an admirable reinforcement for CMCs due to their improved properties and ease of fabrication by the recent innovations in the polymer fibre fabrication technique [2]. Over the last three decades, silicon-based advanced ceramics with a variety of desirable properties, such as remarkable chemical resistance, thermal stability and mechanical strength, have been designed with the help of straightforward polymer-to-ceramic conversion [3,4]. The mechanical properties of PDCFs have been reported as high as 2.5 GPa in tensile strength with 300 GPa in modulus and thermal stability up to 2200°C [5].

With the rising demand for better processability, higher ceramic yield and enhanced properties of the final ceramic product, a number of new silicon-based pre-ceramic polymers were synthesized in the 1980s with modified properties that are sufficient enough for fibre spinning [6–9]. The large-scale production of ceramic fibres started as early as the 1990s [10]. Either melt- or dry-spinning was applied to mainly polysilanes or polyborosilazanes to produce non-oxide SiC or SiBCN fibres. In terms of melt-spinning, modified polymer with required viscosity is continuously fed through a heated (about 150°C) nozzle with hundreds of holes. Fibres are effectively drawn in this way and cured by additional thermal or chemical treatments [10]. Up until recently, there are many types of PDCFs developed over 20 years ago and still commercially available from various manufacturers, such as Hi-Nicalon (from COI), Sylramic (from UBE), etc.

Currently, both oxide and non-oxide (from polymer pyrolysis route) ceramic fibres are widely applied for high-temperature applications. Oxide fibres usually deliver higher oxidation resistance compared with non-oxide fibres, while better creep resistance and strength retention allow non-oxide fibres to be even more suitable for ultra-high temperature applications, such as aerospace components. However, polycarbosilane, the main precursor of SiC fibres, is still costly and limited in availability. Hence, ceramic fibres with considerably lower cost and slightly compromised properties are of great interest [11–13]. Recently, other Si-based ceramics by PDC routes, such as the ternary silicon carbonitride SiCN, synthesized via a cost-effective technique, have been shown to be stable up to 1300°C [14]. The formation of $SiO_2$ and $Si_2N_2O$ double-layered structure generates increased diffusion barrier, thus enhanced oxidation resistance [15,16]. Consequently, pre-ceramic polymers with higher N content may yield PDCs with improved thermal and oxidation stability of the final ceramic product.

Here, we report a new approach to synthesize carbon-rich SiOCN ceramic fibres via hand-drawing and pyrolysis of a cyclic-silazane/polyacrylic acid (PAA) hybrid polymer. The hybrid approach, involving hand-spinning process, is highly efficient for laboratory-scale testing and to develop fundamental understanding related to the processing of pre-ceramic fibres and related polymer to ceramic transformations without the need for complex fibre-drawing equipment. 1,3,5,7-tetramethyl-1,3,5,7-tetravinyl-cyclotetrasilazane (referred as TTCSZ hereafter) is an organosilazane, whose properties have not been reported although its siloxane analogue, 1,3,5,7-tetramethyl-1,3,5,7-tetravinyl cyclotetrasiloxane is a common precursor for SiOC ceramic. However, this type of silazane is cross-linkable at elevated temperature, but it is not known to form fibres in its neat form. Hence, PAA is introduced to improve the spinnability of pre-ceramic polymer. As-prepared fibres were firstly cross-linked at lower temperature in open air, then pyrolysed at higher temperature for polymer-to-ceramic conversion under the protection of argon. Synthesis, characterization, pyrolysis and phase formation are discussed in detail for TTCSZ/PAA hybrid fibres in the following sections.

# 2. Experimental

## 2.1. Materials

All syntheses were conducted with as-received materials without further treatment or purification. The TTCSZ is available from Gelest (USA). The PAA is purchased from Sigma–Aldrich (USA). Dicumyl peroxide (DCP) as cross-linking initiator is obtained from Sigma–Aldrich (USA). The fibre drawing, handling and cross-linking processes are performed in open air, except pyrolysis which is carried out in ultra-high purity argon atmosphere from Matheson (USA).

## 2.2. Fibre drawing and pyrolysis

TTCSZ/PAA hybrid fibres were prepared via the sol–gel process (figure 1). Initially, 10 wt% of DCP dissolved in TTCSZ formed a solution. PAA was initially mixed with deionized water (PAA to de-ionized (DI) water wt ratio 4 : 1) and stirred until the mixture turned completely into transparent gel. TTCSZ solution was then added to the gel (TTCSZ solution to PAA wt ratio 1:1). After complete mixing by stirring, the PAA/TTCSZ/water system formed a piece of white slurry with a viscosity of about 250 P (poise) or 25 Pa s. The overall ratio between TTCSZ : DCP : PAA was 45 : 5 : 50 wt%. The fibre-drawing process is carried out by hand-drawing in open air. The hybrid fibres can be conveniently drawn directly from the slurry using metal rods, spatulas or tweezers onto aluminium boats for cross-linking process. Then the fibres are dried and cross-linked in a low-temperature oven at 160°C for 24 h in air. The aluminium boats are simply hand-crafted with aluminium foils which have several advantages to be applied in this process. The aluminium foil is chemically and mechanically stable as a supporter that does not introduce extra impurities in low-temperature open-air environment; the shrinkage of the hybrid fibres during cross-linking process can be well controlled by the boat; the size of the boat can be determined by the size of the oven. Fibres are collected from aluminium boat after cross-linking, then relocated and wrapped on a small boat or rod support made with ceramics or quartz for high-temperature pyrolysis.

The polymer-to-ceramic transformation is performed at three different temperatures (600, 700 and 800°C). The fibres were heated from room temperature to target temperature at a heating rate of 2°C min$^{-1}$ in argon (figure 2).

## 2.3. Characterization

The surface morphology of the fibres at each stage was investigated by a scanning electron microscopy (EVO MA10, ZEISS, Germany).

Raman spectroscopy was collected from LabRAM ARAMIS Raman spectrometer (LabRAM HORIBA Jobin Yvon, USA) using a HeNe laser (633 nm) as an excitation source.

Fourier-transform infrared spectroscopy (FT-IR) spectra were obtained from Spectrum 400 FT-IR spectrometer (Perkin Elmer, Waltham, MA) on a diamond crystal top plate of an ATR accessory (GladiATR, Pike Technologies, USA).

Viscosity measurement was performed on a Brookfield DV-II+Pro viscometer (Brookfield Engineering, USA) using CEP-40 and CEP-52 cones for neat TTCSZ and TTCSZ/PAA, respectively; 0.5 ml slurry was used with 25–200 r.p.m. rotation rate at 37°C for the measurement.

X-ray photoelectron spectroscopy (XPS) was performed by PHI Quantera SXM (ULVAC-PHI, Japan). Depth-profiling was applied for 15 min via Ar ion-beam.

Solid-state $^{13}$C MAS, CP MAS and $^{29}$Si MAS NMR spectra were recorded on a Bruker AVANCE 300 spectrometer ($B_0 = 7.0$ T, $v_0(^1$H$) = 300.29$ MHz, $v_0(^{13}$C$) = 75.51$ MHz, $v_0(^{29}$Si$) = 59.66$ MHz) using either 4 or 7 mm Bruker probes and spinning frequencies of 10 or 5 kHz. $^{13}$C CP MAS experiments were recorded with ramped-amplitude cross-polarization in the $^1$H channel to transfer magnetization from $^1$H to $^{13}$C (recycle delay = 3 s, CP contact time = 1 ms, optimized $^1$H spinal-64 decoupling). Single-pulse $^{13}$C and $^{29}$Si NMR MAS spectra were recorded with recycle delays of 30 and 60 s, respectively. Chemical shift values were referenced to TTCSZ for $^{13}$C and $^{29}$Si.

The fibre thermal resistance was analysed using thermal gravimetric analysis (TGA). The analysis was performed with TG 209 F1 Libra (NETZSCH, Germany) from 100 to 1000°C (with a heating rate of 10°C min$^{-1}$) in either N$_2$ or synthetic air (with a flow of 10 ml min$^{-1}$) for cross-linked, pyrolysed TTCSZ/PAA and neat PAA samples.

# 3. Results and discussion

## 3.1. Characterization of the hybrid fibres

A series of characterization was carried out to determine the properties of as-pyrolysed hybrid PDC fibres. SEM images of the pyrolysed hybrid fibres are given in figure 3a–c. It is confirmed by SEM that hand-spun fibres have a diameter range of 1–10 µm. Each pyrolysed fibre is considerably uniform in diameter (along the length) and porous at the surface.

The chemical compositions of PDC hybrid fibres are determined by XPS depth-profiling survey scans and the results are listed in table 1. The presence of oxygen and loss of nitrogen are discussed in the

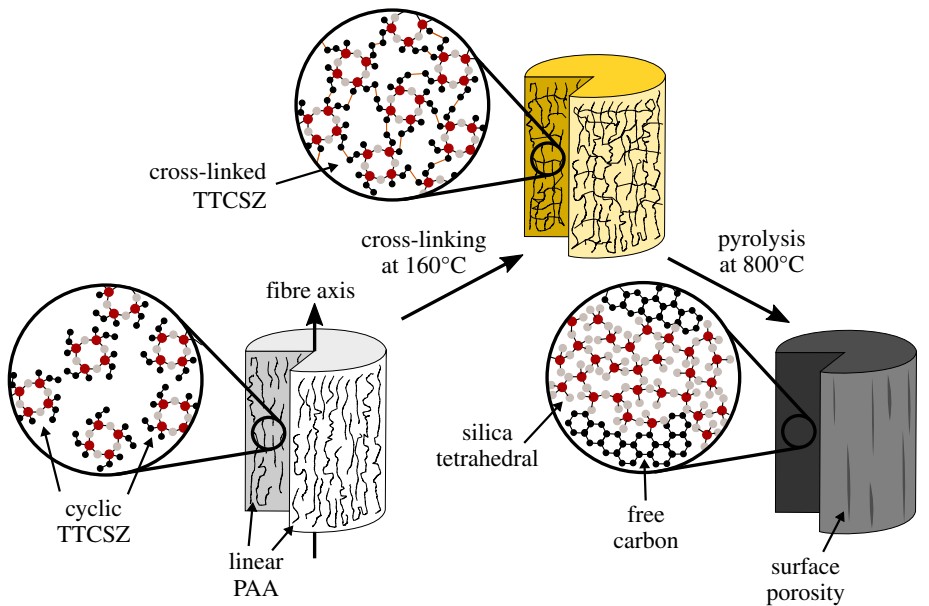

**Figure 1.** Schematics of fibre-drawing process.

**Figure 2.** Schematic showing fibre cross-linking and pyrolysis process. TTCSZ is the main PDC fibre precursor where PAA plays a fibre template role in the hybrid system. TTCSZ molecules are embedded in PAA rigid environment that can be cross-linked at an elevated temperature. Within the fibres, PDC nano-domain structures are formed after being pyrolysed at even higher temperature (≧ 800°C).

following sections. The existence of 'free carbon' was confirmed by the Raman spectroscopy. Carbon G-band (approx. 1595 cm$^{-1}$) (figure 4d,e) shows the existence of graphitic sp$^2$ carbon structure in the fibres, while D-band (approx. 1340 cm$^{-1}$) represents a large amount of disordered carbon [17].

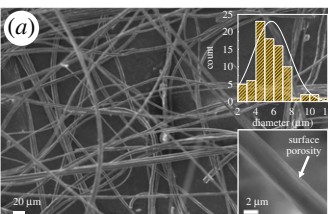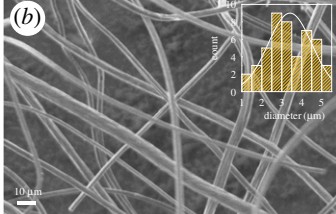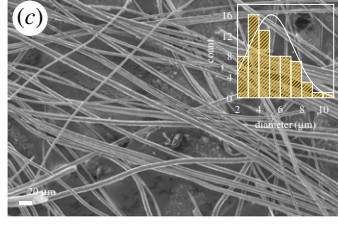

**Figure 3.** SEM images of the hand-spun TTCSZ/PAA hybrid fibres pyrolysed at (*a*) 600°C, (*b*) 700°C and (*c*) 800°C. Fibres have a diameter distribution of 1–10 μm with an average diameter about 5 μm. When magnified, fibre surface porosity is notable.

**Table 1.** Chemical composition of fibres by XPS depth-profiling survey scan in atomic percentage.

| pyrolysis temperature (°C) | element atomic percentage (at. %) | | | |
|---|---|---|---|---|
| | C | Si | O | N |
| 700 | 71.2 | 8.6 | 19.4 | 0.3 |
| 800 | 50.8 | 15.5 | 33.4 | 0.4 |

Specifically, D4- and D3-bands indicate the existence of $sp^2$-$sp^3$ bonds and amorphous carbon in the fibres [17,18]. Carbon peaks are relatively weaker (high fluorescence) for the fibres pyrolysed at 600°C indicating less free carbon. For the higher pyrolysis temperature, free carbon amount is significantly increased. Correspondingly, graphitic carbon is increased with the increase of pyrolysis temperature, while the amount of amorphous carbon is decreased. Because Si–C bonds are $sp^3$, stronger D1, D4-bands in 700°C sample confirms higher $SiO_2C_2$ and $SiO_3C$ composition discussed below with the NMR analysis (figure 5*a*) [19]. Raman spectra of neat TTCSZ samples are also obtained to find out the contribution of PAA to the pyrolysed fibres (figure 4*b*). The free carbon amounts are apparently raised with the incorporation of PAA at initial stage by comparing the intensity of the carbon peaks. Further analysis was performed on bonding types and molecular structures. NMR spectroscopy investigation was performed on the samples at each pyrolysis temperature (figure 5). $^{29}$Si MAS NMR spectrum (figure 5*a*) of the sample heat-treated at 600°C shows components at approximately −20 ppm, approximately −65 ppm and approximately −110 ppm, suggesting tetrahedrally coordinated Si in $SiO_2C_2$, $SiO_3C$ and $SiO_4$ species typical of mixed silicon oxycarbide units [20,21]. It should be noted that part of the signal at approximately −10 ppm could also correspond to $SiN_2C_2$ environments [22] but probably in small proportions. This point will be further discussed in §3.3. The strong intensity of $SiO_3C$ signal indicates a significant amount of C bonded to Si atoms and correspondingly less free carbon, this result well matches with the Raman data. When increasing the pyrolysis temperatures, the fractions of mixed silicon oxycarbide units decreases: $SiO_3C$ peak greatly decreases and $SiO_2C_2$ peak disappears completely. $SiO_3C$ peak further decreases at 800°C and leaving most Si atoms in silica tetrahedral form.

$^{13}$C CP MAS NMR results (figure 5*b*) show a clear free carbon peak around 130 ppm for both 600 and 700°C. No remaining vinyl groups are observed after pyrolysis, suggesting a complete polymerization into aliphatic carbons, while a strong Si–C peak is detected at 600°C which drastically decreases at 700°C.

Furthermore, NMR spectra were recorded from TTCSZ/PAA hybrid fibres and neat TTCSZ powder samples pyrolysed at 700°C (figure 6) to investigate the effect of PAA on the structure of the final product. From the comparison, neat TTCSZ at 700°C shows a spectrum very close to TTCSZ/PAA at 600°C. As it has been already shown in Raman analysis, either incorporation of PAA or a rise in pyrolysis temperature will ultimately result in a structure with elevated 'free carbon' content and decreased Si-C bonds. This is also confirmed by $^{13}$C CP MAS NMR data showing increased 'free carbon' content and decreased Si–$CH_x$ signals in TTCSZ/PAA hybrid sample compared with neat TTCSZ.

XPS analysis was applied to determine the bonding type (figure 7) from survey scan and high-resolution scan of O1s, C1s, Si2s, Si2p and N1s peaks in each sample [23]. From the survey scan, N1s peak disappears after pyrolysis above 600°C. High-resolution scans of major peaks reveal the merging of different signals, such as Si–C peaks at approximately 102.0 eV in Si2p scan and approximately 284.6 eV in C1s scan [23]. $SiO_2C_2$, $SiO_3C$ and $SiO_4$ components can be obtained from hybrid fibres pyrolysed at 600°C in agreement with the NMR results [24]. From high-resolution C1s scan

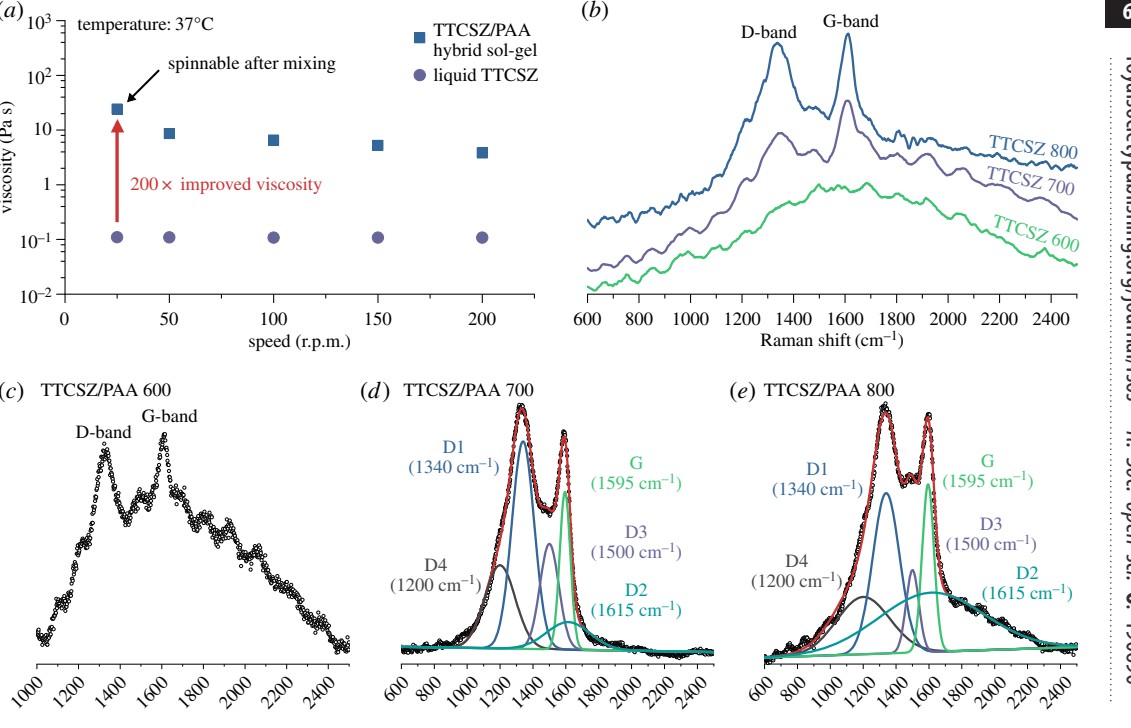

**Figure 4.** (*a*) Viscosity of TTCSZ/PAA slurry and liquid TTCSZ; 200 times improved viscosity is observed after mixing with PAA, which enhanced spinnability. (*b*) Raman spectroscopy of TTCSZ pyrolysed at different temperatures. TTCSZ/PAA hybrid fibres pyrolysed at (*c*) 600°C, (*d*) 700°C and (*e*) 800°C. Free carbon amount is increased as the pyrolysis temperature is raised.

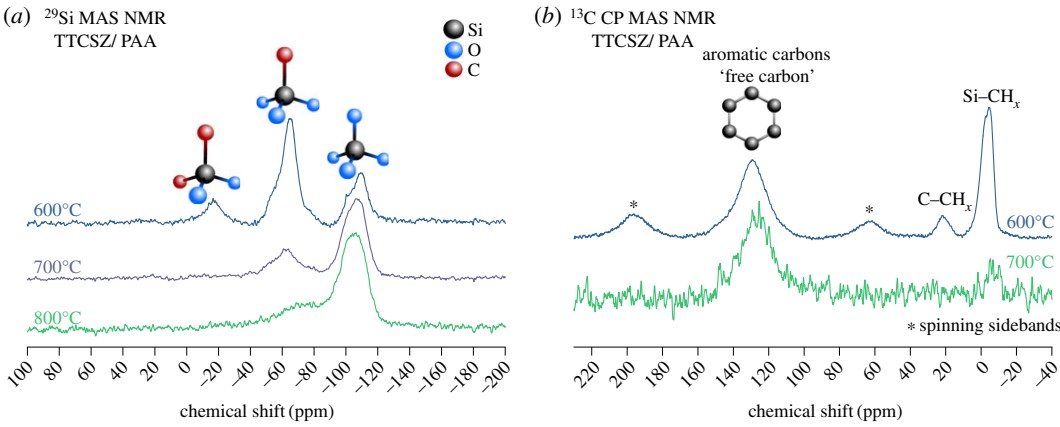

**Figure 5.** Solid-state NMR spectra of pyrolysed TTCSZ/PAA hybrid fibres at 600, 700 and 800°C (*a*) $^{29}$Si MAS (*b*) $^{13}$C CP MAS with corresponding microstructure shown. A significant drop in Si–C bonds is noted as pyrolysis temperature is increased. Number of scans (NS): $^{29}$Si MAS 600°C (NS = 1334), 700°C (NS = 1122), 800°C (NS = 4186); $^{13}$C CP MAS 600°C (NS = 2400), 700°C (NS = 2784).

(figure 7*a*.iii, *b*.iii, *c*.iii), the C–Si component is continuously decreased leading to the relative increase in C–C as the pyrolysis temperature increases. This result also matches the FT-IR spectra in figure 8.

## 3.2. Spinnability

Initial attempts were made to draw fibres directly from neat liquid TTCSZ. The rapid cross-linking was observed at about 125°C with 10 wt% cross-linking initiator DCP and completely transformed TTCSZ liquid into a hard bulk, without an intermediate viscous status that can initiate the spinning process. The failure was due to two main reasons: low viscosity (figure 4*a*) and cyclic structure of TTCSZ. PAA was introduced to the initial fibre-drawing stage before cross-linking to improve the spinnability

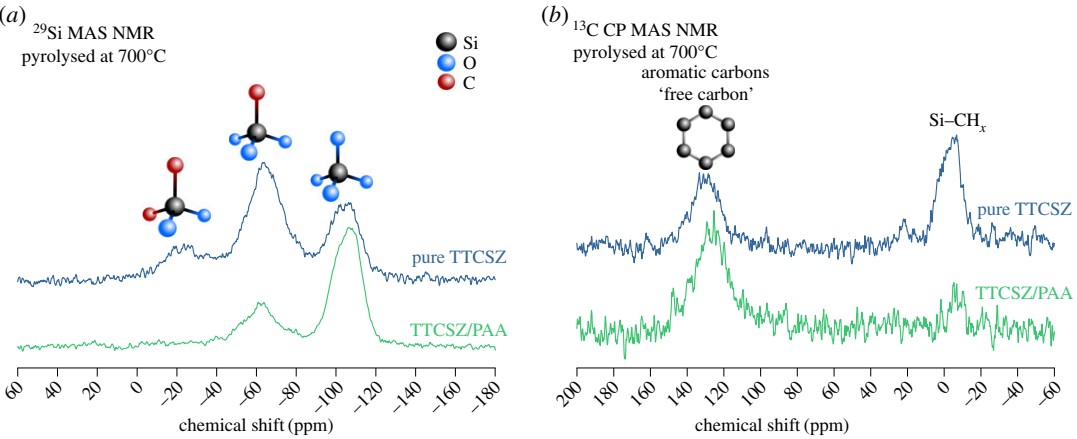

**Figure 6.** Solid-state NMR spectra of TTCSZ/PAA fibres and neat TTCSZ at 700°C (a) $^{29}$Si MAS (b) $^{13}$C CP MAS. Under the same pyrolysis temperature, incorporation of PAA leads to increased free carbon and decreased Si–C bonds. Number of scans (NS): $^{29}$Si MAS neat TTCSZ (NS = 1396), TTCSZ/PAA (NS = 1122); $^{13}$C CP MAS neat TTCSZ (NS = 2736), TTCSZ/PAA (NS = 2784).

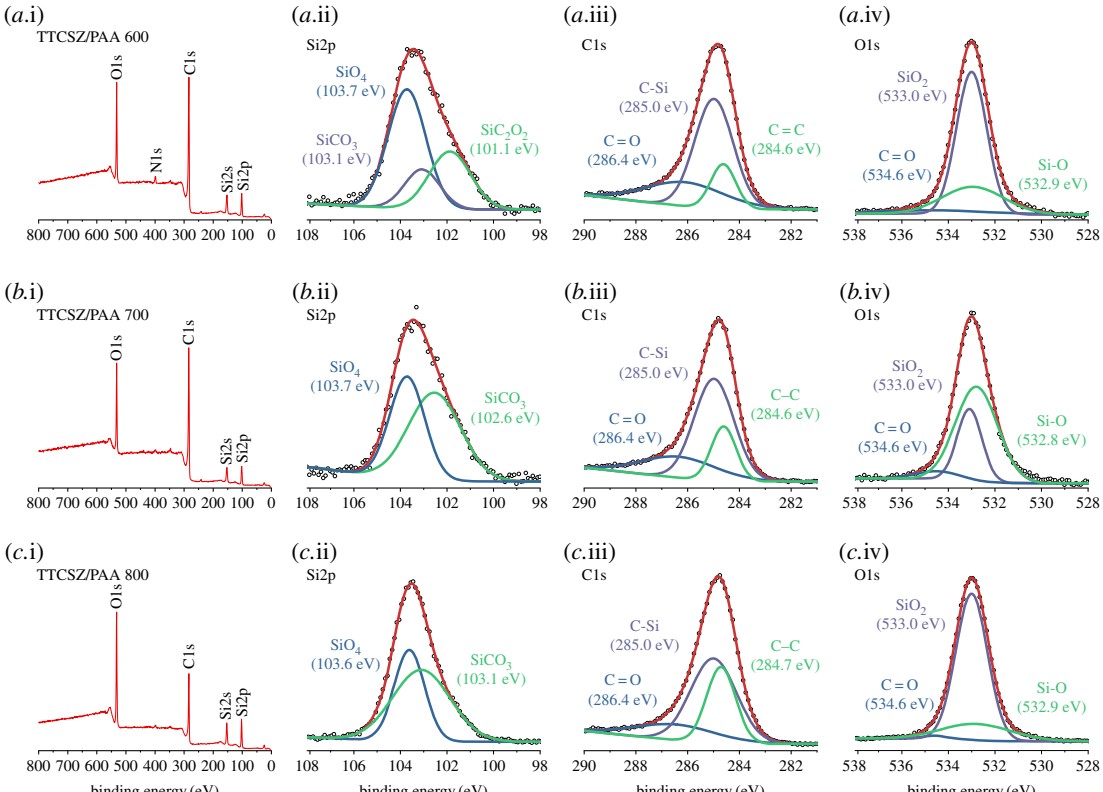

**Figure 7.** XPS results of PDC fibres from TTCSZ/PAA hybrid precursor. (a–c.i) Survey scan. High-resolution scan of Si2p, C1s and O1s of samples pyrolysed at (a.ii–iv) 600°C, (b.ii–iv) 700°C and (c.ii–iv) 800°C. From (a.iii) to (c.iii), there is a huge increase in C–C/C–Si ratio.

of TTCSZ. Viscosity measurement showed low viscosity (approx. 0.1 Pa s at 37°C) for liquid TTCSZ with 10 wt% DCP. After mixing with PAA, viscosity was raised more than 200 times to about 25 Pa s at a shear rate of 25 r.p.m. (or 50 cm s$^{-1}$) which met the requirement for sol–gel spinning (greater than 10 P) [25]. Besides, $^{13}$C CP MAS NMR (figure 9) shows broad signals (suggesting low mobility) of –CH(COOH)–CH$_2$– species which is responsible for enhanced spinnability.

## 3.3. Cross-linking and pyrolysis behaviour

Investigation of the cross-linking behaviour is essential since the final content of PDC fibres is largely determined by the bonds formed during cross-linking stage (figure 2). FT-IR analysis was obtained with main band characteristics of raw PAA fibres [26–28], neat PAA fibres heated at 160°C, and

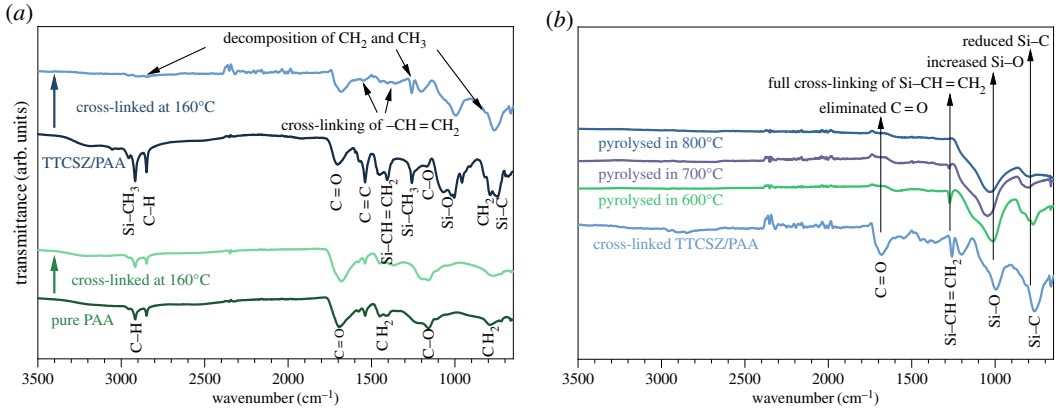

**Figure 8.** FT-IR spectra of TTCSZ/PAA hybrid fibres (*a*) cross-linked at 160°C, (*b*) pyrolysed at 600, 700 and 800°C. At cross-linking stage, decomposition of CH$_2$ and CH$_3$, crosslinking of −CH = CH$_2$ are clearly observed.

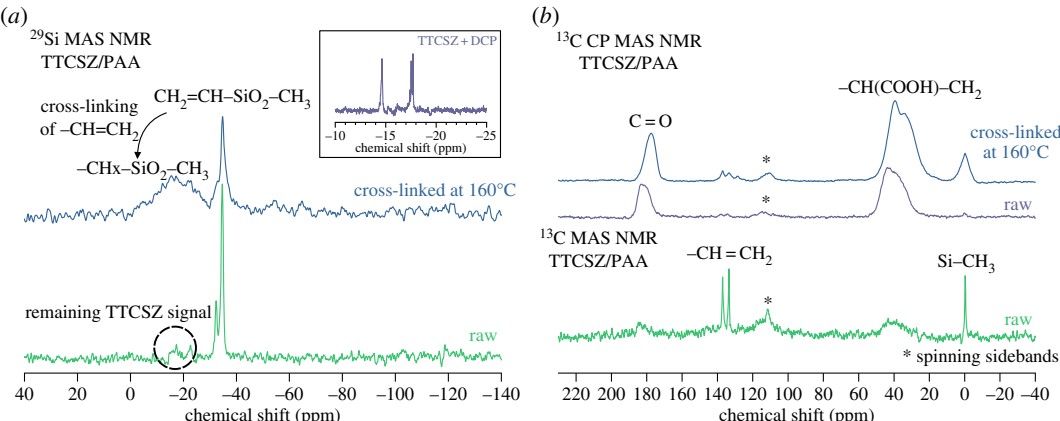

**Figure 9.** Solid-state NMR spectra of TTCSZ/PAA hybrid raw fibres and cross-linked fibres. $^{29}$Si MAS NMR result indicates decreased mobility after cross-linking, and $^{13}$C showed very low interaction between PAA (rigid environment) and TTCSZ at cross-linking stage. Number of scans (NS): $^{29}$Si MAS cross-linked at 160°C (NS = 474), raw (NS = 140); $^{13}$C CP MAS cross-linked at 160°C (NS = 1056), raw (NS = 432); $^{13}$C MAS raw (NS = 56).

TTCSZ/PAA hybrid raw, cross-linked and pyrolysed fibres [29,30]. The weak influence of DCP on FT-IR signal is discussed in electronic supplementary material, figure S1. FT-IR is mainly used to investigate the cross-linking behaviour of the precursors [29]. The changes between the precursor spectra and the cross-linked spectra reveal a critical loss of hydrogen and nitrogen during cross-linking. The excessive oxygen content is possibly imported by the initiator, adsorbed from the air or introduced during the reaction with water (discussed below). Moreover, the majority of organic moieties (methyl and vinyl groups) are eliminated during this process along with the cross-linking evidence shown in figure 8*a*.

NMR on raw and cross-linked samples supports FT-IR results in figure 9. Both $^{13}$C MAS and $^{13}$C CP MAS NMR were obtained for raw fibres to determine, respectively, highly mobile components (TTCSZ) and rigid environments (PAA). Before cross-linking, both $^{29}$Si and $^{13}$C MAS spectra show narrow signals, indicating high-mobility species: $^{29}$Si peaks are mainly observed around −35 ppm, a position significantly different from the initial TTCSZ+DCP solution showing signals at −14.7 ppm and −17.5 ppm (figure 9*a*, inset) characteristic of 6-ring 2,4,6-trimethyl-2,4,6-trivinyl-cyclotrisilazane [31] and 8-ring silazane TTCSZ, respectively. Signals around − 33 ppm are indeed reported for 1,3,5,7-tetravinyl-1,3,5,7-tetracyclotetrasiloxane [32]. This strongly suggests a significant oxidation of the silazane rings into siloxanes, possibly due to the addition of water with PAA. Nonetheless, additional minor signals are still present around −17 ppm, indicating that silazane bonds are still present.

At this point, from the strong evidence obtained from NMR analysis, two different hypotheses are proposed here to explain the loss of nitrogen before pyrolysis:

(1) A large amount of the nitrogen loss may be addressed to the pre-spinning mixing stage. −NH− may react with H$_2$O molecules and form NH$_3$ inducing nitrogen loss before cross-linking.

$$\underset{\text{(TTCSZ structure)}}{\cdots} + H_2O \longrightarrow \underset{\text{(oxidized structure)}}{\cdots} + NH_3$$

(2) During the cross-linking stage, the hydrosilylation reaction takes place. This reaction is induced by heating and happens at relative lower temperature (as low as 120°C) [33,34].

$$2 \equiv Si-NH-Si \equiv \longrightarrow \equiv Si-N\Big\langle {}^{Si\equiv}_{Si\equiv} + \equiv Si-NH_2$$

$$\equiv Si-NH_2 + \equiv Si-NH-Si \equiv \longrightarrow \equiv Si-N\Big\langle {}^{Si\equiv}_{Si\equiv} + NH_3$$

As a result, TTCSZ may be oxidized to 1,3,5,7-tetramethyl-1,3,5,7-tetravinyl-cyclotetrasiloxane. Then methyl-vinyl radical reaction will take place initiated by DCP during cross-linking stage. The reaction can be shown as follows [35].

$$\equiv Si-CH=CH_2 + \equiv Si-CH_3 \xrightarrow{\text{DCP·O·}} \equiv Si \diagdown \diagup Si \equiv \text{ or } \equiv Si \diagup Si \equiv$$

The reactions are expected to take place on all four vinyl groups attached to four cyclic silicon atoms and form into a network structure. However, this cross-linking reaction may not be complete due to the limited mobility after the network formation [34,35].

$$\text{(cyclic siloxane)} \xrightarrow{\text{DCP·O·}} [\cdots]_a + [\cdots]_b + [\cdots]_c + [\cdots]_d$$

After cross-linking at 160°C, broader peaks indicate less mobility due to the conversion of Vi–(Me)SiO$_2$ to –CH$_2$-(Me)SiO$_2$. However, high consistency between raw and 160°C $^{13}$C CP MAS suggests very weak PAA cross-linking activities at 160°C. Further discussion regarding cross-linking behaviour between PAA and TTCSZ is presented in electronic supplementary material, figure S2.

Polymer-to-ceramic conversion process is revealed at 600°C which is supported by Raman analysis showing free carbon bands in figure 3. However, the comparison of FT-IR results between cross-linked fibres and pyrolysed fibres suggests that the conversion is not complete until 800°C. The decomposition of remaining organic groups is observed at pyrolysis temperature range 600–800°C.

## 3.4. Thermal stability

The thermal behaviour of the hybrid fibres is studied with thermogravimetric analysis using powder samples in either N$_2$ or air and the results are shown in figure 10. The significant weight loss of neat PAA fibres started at a temperature as low as 200°C. The hybrid fibres pyrolysed at 800°C showed a good thermal stability up to 1000°C with the weight loss of 4.5 wt%, while the TGA curve of cross-linked hybrid samples gives an estimation of ceramic yield of this TTCSZ/PAA combination. The estimated yield of cross-linked sample is about 48% according to figure 10, which well matches with the recorded ceramic yield. TGA result of neat PAA in air, associated with NMR results, indicates that PAA in the hybrid system mainly acts as a fibre template (rigid environment mentioned above) in the spinning process and does not participate in cross-linking ($^{13}$C CP MAS NMR) and provides a small amount of excess carbon after pyrolysis.

# 4. Conclusion

Carbon-rich SiOCN fibres are synthesized and reported in this work from commercially available 1,3,5,7-tetramethyl-1,3,5,7-tetravinyl-cyclotetrasilazane and PAA forming a hybrid pre-ceramic polymer

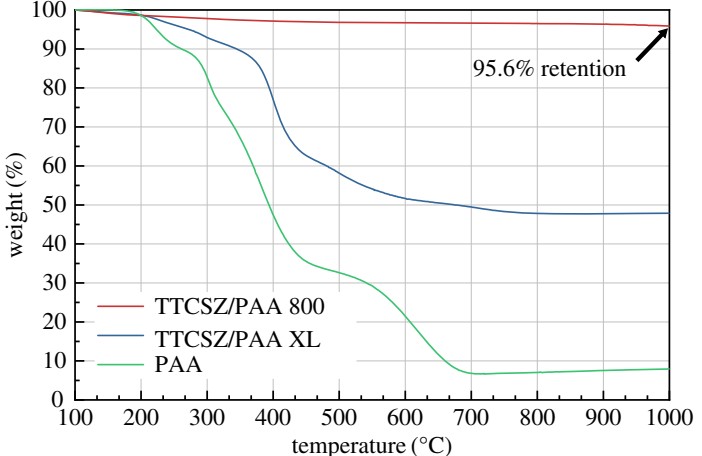

**Figure 10.** TGA analysis of TTCSZ/PAA hybrid pyrolysed at 800°C and weight change recorded in N$_2$; TTCSZ/PAA hybrid cross-linked at 160°C and weight change recorded in N$_2$; neat PAA sample and weight change recorded in air.

precursor. The sol–gel process, fibre-drawing and cross-linking processes are easily conducted in open air. Linear PAA molecules provide rigid body during fibre-drawing process and provide excess free carbon in the final fibre product. As-spun fibres are revealed oxygen-rich, which may be due to the presence of water during mixing stage, still, providing intrinsic thermal stability at 1000°C.

The fibre products are uniform in diameter even with hand-spinning. A small degree of surface porosity (without large defects) observed from the fibres during the characterization (BET surface analysis data presented in electronic supplementary material, figure S3). The spinning process and technique developed in this work successfully produced fibres from an unspinnable silazane with acceptable compromise in composition and properties, but greatly reduced processing cost and environmental requirement. This hybrid technique is recommended to laboratory-scale early production of new types of ceramic fibres. High productivity of this technique can easily satisfy various characterization needs with minimum set-up requirement. The fibre products are suggested as reinforcement for CMC materials.

The fibre structure and properties may be improved by altering synthesis condition. Oxygen content can be controlled from the initial fibre-drawing stage to cross-linking stage: instead of mixing TTCSZ/ PAA in the aqueous environment, melt-spinning is applicable to avoid the reaction between TTCSZ and water molecule, where this process may be conducted in a glovebox; cross-linking environment may be adjusted to inert atmosphere such as N$_2$ or Ar to avoid fibres having direct contact with O$_2$; cross-linking method may also be altered to irradiation (gamma or electron), photo (UV) or ionic cross-linking instead of thermal cross-linking.

Data accessibility. Raw data are available within the Dryad Digital Repository at: https://doi.org/10.5061/dryad.bt45780 [36].

Authors' contributions. G.S. participated in design of the project as a senior researcher and helped with drafting the manuscript; C.G. contributed to perform characterizations, data analysis and drafted the manuscript; Z.R. carried out the preparation of all samples, data analyses and drafted the manuscript.

Competing interests. KSURF Disc. 2019-027. Title: 'SiCNO Ceramic Fibers from Hybrid Organic/inorganic Preceramic Polymer'. July 2019. (Invention disclosure).

Funding. Financial support from National Science Foundation grant no. 1743701 is gratefully acknowledged.

Acknowledgements. C.G. thanks F. Ribot for the fruitful discussions. G.S. and Z.R. thank undergraduates MacKenzy Meis and Isabella Cesarone for their help with the material preparation. The authors thank especially Dr Qiang Ye from Institute for Bioengineering Research Laboratories, University of Kansas for equipment (Raman, FT-IR and viscometer) training and technical support.

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
