## [Reviewer comments · Royal Society Open Science]

Review History

RSOS-190690.R0 (Original submission)

Review form: Reviewer 1 (Chrystelle Salameh)

Is the manuscript scientifically sound in its present form?

Yes

Are the interpretations and conclusions justified by the results?

Yes

Is the language acceptable?

Yes

Is it clear how to access all supporting data?

Yes

Do you have any ethical concerns with this paper?

No

Have you any concerns about statistical analyses in this paper?

No

Recommendation?

Accept with minor revision (please list in comments)

Comments to the Author(s)

The MS by Ren et al entitled "Preparation and structure of SiOCN Fibers Derived from Cyclic Silazane/PAA Hybrid Precursor" deals with an easy and cost-effective approach to fabricate Si-based ceramic fibers by simple hand spinning of a hybrid preceramic polymer composed of a commercial silazane and PolyAcrylic Acid (PAA) as an additive to allow the direct spinning of the polymers. This topic is very interesting for the High-temperature applications (turbines, aerospace materials...) and the process investigated by the authors is a very good contribution to the field and might be of great interest for the industries since it can be easily upscaled and manufactured.

The research is well conducted and the MS is nicely written. However minor topological errors were found, authors are encouraged to correct for example in the introduction (page 2, line 37) "drawing" instead of "drwaing". In table 1 (page 4) "Si" instead of "S" in the column related to the combustion analysis...

A detailed characterization of the hybrid and ceramic fibers was conducted with a special attention on the solid-state NMR which is the "technique of choice" to understand in details the structure of such systems as well as their evolution with the temperature. Another information offered by this technique was the spinnability of the polymers. The authors succeeded thus in optimizing the viscosity of the hybrid polymer by adjusting the ratio between the PAA and the silazane in order to successfully "hand-draw" the synthesized hybrid gel.

The elemental analysis performed on the hybrid fibers are shown at 2 different pyrolysis temperatures in Table 1. The N content is clearly low and keeps on decreasing with the temperature increase. Even though the authors explained the loss of nitrogen in page 9 paragraph C, which I agree with, the elemental composition presented describes more a SiOC system rather than SiOCN. I suggest that authors include the N content in their final formula if they want to describe their ceramic as SiOCN. I also suggest that they put the NMR signals related to silazanes directly on the spectra to support their argument stated in page 9, line 24: "nonetheless additional minor signals are still present around -17ppm, indicating that silazane bonds are still present". It will be easier for the reader to keep track.

Based on the aforementioned remarks I recommend this MS for publication.

Review form: Reviewer 2**Is the manuscript scientifically sound in its present form?**

No

Are the interpretations and conclusions justified by the results?

Yes

Is the language acceptable?

Yes

Is it clear how to access all supporting data?

Not Applicable

Do you have any ethical concerns with this paper?

No

Have you any concerns about statistical analyses in this paper?

No

Recommendation?

Major revision is needed (please make suggestions in comments)

Comments to the Author(s)

The paper deals with the synthesis optimization of SiOCN fibers made starting from cyclic silazanes and polyacrylic acid, which seems to act as templating agent during thermal treatment. The process is described in details and the possible reactions that take place during thermal treatments, too. Several analytical techniques such as SEM, IR, XPS and solid state NMR have been used to elucidate the different stages of the materials. TG has been used for stability evaluation.

The work is described in details and every step is adequately commented and proved with the above-mentioned techniques.

However, there are some points that must be clarified to give value to the work, especially dealing with the reason of the reagent choice that is a big part.

* Introduction: there is the need of more references about CMC and fiber production (see, just for example, Hiroyuki Takeuchi, Kaneo Noake, Tamio Serita's patent). A lot of work has been already done.

*Characterization: number of scans for NMR spectra?

*Results

- Si instead of S, in table 1.

- It is not clear why Si will reduces at high pyrolytic temperatures

- It is declared that from Raman a lot of free C is present, thus it cannot be taken into account in the proposed formula in table 1

- At least in two part of the paper the porosity of the fiber is mentioned. It should be a good idea to measure N₂ physisorption in order to better define this porosity.

- The amount of the DCP seems very high, so that should be considered as a "reagent". Could the author explain the reason for this choice? Is it detectable with NMR or FTIR? Is it washed away?

- Although the chemical processes are well described and commented, and the final properties of the materials seems valuable, the processes result critical for various reason: as stated above the amount of the radical initiator, the usefulness of the silazane to produce materials with no or negligible amount of N in the structure, which, moreover, causes the release of NH₃. This by-product, on an industrial scale, will be a problem due to its toxicity.

Review form: Reviewer 3 (Zhaoju Yu)

Is the manuscript scientifically sound in its present form?

Yes

Are the interpretations and conclusions justified by the results?

Yes

Is the language acceptable?

Yes

Is it clear how to access all supporting data?

Yes

Do you have any ethical concerns with this paper?

No

Have you any concerns about statistical analyses in this paper?

No

Recommendation?

Accept with minor revision (please list in comments)

Comments to the Author(s)

This paper reported Preparation and Structure of SiOCN Fibers Derived from Cyclic Silazane/PAA Hybrid Precursor, which fits the scope of the journal Royal Society Open Science. The paper is straightforward and well-organized. I would recommend publication of this work after minor revision. The introduction of PAA to TTCSZ to create a hybrid precursor remarkably improved spinnability of the silazane and should be widely applicable to other unspinnable PDC preceramic polymers. What happened between the PAA and TTCSZ? Was there chemical reaction involved? Can the authors show the proof and comment more on this issue?

Decision letter (RSOS-190690.R0)

03-Jun-2019

Dear Mr Ren,

The editors assigned to your paper ("Preparation and Structure of SiOCN Fibers Derived from Cyclic Silazane/PAA Hybrid Precursor") have now received comments from reviewers. We would like you to revise your paper in accordance with the referee and Associate Editor suggestions which can be found below (not including confidential reports to the Editor). Please note this decision does not guarantee eventual acceptance.

Please submit a copy of your revised paper before 26-Jun-2019. Please note that the revision deadline will expire at 00.00am on this date. If we do not hear from you within this time then it will be assumed that the paper has been withdrawn. In exceptional circumstances, extensions may be possible if agreed with the Editorial Office in advance. We do not allow multiple rounds of revision so we urge you to make every effort to fully address all of the comments at this stage. If deemed necessary by the Editors, your manuscript will be sent back to one or more of the original reviewers for assessment. If the original reviewers are not available, we may invite new reviewers.

When submitting your revised manuscript, you must respond to the comments made by the

referees and upload a file "Response to Referees" in "Section 6 - File Upload". Please use this to document how you have responded to the comments, and the adjustments you have made. In order to expedite the processing of the revised manuscript, please be as specific as possible in your response.

- Data accessibility

If you wish to submit your supporting data or code to Dryad (<http://datadryad.org/>), or modify your current submission to dryad, please use the following link:
<http://datadryad.org/submit?journalID=RSOS&manu=RSOS-190690>

- Competing interests

- Authors' contributions

- Acknowledgements

- Funding statement

Kind regards,

Alice Power

Editorial Coordinator

on behalf of Dr Maria Charalambides (Associate Editor) and R. Kerry Rowe (Subject Editor)

Comments to Author:

Reviewers' Comments to Author:

Reviewer: 1

Comments to the Author(s)

The MS by Ren et al entitled "Preparation and structure of SiOCN Fibers Derived from Cyclic Silazane/PAA Hybrid Precursor" deals with an easy and cost-effective approach to fabricate Si-based ceramic fibers by simple hand spinning of a hybrid preceramic polymer composed of a commercial silazane and PolyAcrylic Acid (PAA) as an additive to allow the direct spinning of the polymers. This topic is very interesting for the High-temperature applications (turbines, aerospace materials...) and the process investigated by the authors is a very good contribution to the field and might be of great interest for the industries since it can be easily upscaled and manufactured.

The research is well conducted and the MS is nicely written. However minor topological errors were found, authors are encouraged to correct for example in the introduction (page 2, line 37) "drawing" instead of "drwaing". In table 1 (page 4) "Si" instead of "S" in the column related to the combustion analysis...

A detailed characterization of the hybrid and ceramic fibers was conducted with a special attention on the solid-state NMR which is the "technique of choice" to understand in details the structure of such systems as well as their evolution with the temperature. Another information offered by this technique was the spinnability of the polymers. The authors succeeded thus in optimizing the viscosity of the hybrid polymer by adjusting the ratio between the PAA and the silazane in order to successfully "hand-draw" the synthesized hybrid gel.

The elemental analysis performed on the hybrid fibers are shown at 2 different pyrolysis temperatures in Table 1. The N content is clearly low and keeps on decreasing with the temperature increase. Even though the authors explained the loss of nitrogen in page 9 paragraph C, which I agree with, the elemental composition presented describes more a SiOC system rather than SiOCN. I suggest that authors include the N content in their final formula if they want to describe their ceramic as SiOCN. I also suggest that they put the NMR signals related to silazanes directly on the spectra to support their argument stated in page 9, line 24: "nonetheless additional minor signals are still present around -17ppm, indicating that silazane bonds are still present". It will be easier for the reader to keep track.

Based on the aforementioned remarks I recommend this MS for publication.

Reviewer: 2

Comments to the Author(s)

The paper deals with the synthesis optimization of SiOCN fibers made starting from cyclic silazanes and polyacrylic acid, which seems to act as templating agent during thermal treatment. The process is described in details and the possible reactions that take place during thermal treatments, too. Several analytical techniques such as SEM, IR, XPS and solid state NMR have been used to elucidate the different stages of the materials. TG has been used for stability evaluation.

The work is described in details and every step is adequately commented and proved with the above-mentioned techniques.

However, there are some points that must be clarified to give value to the work, especially dealing with the reason of the reagent choice that is a big part.

* Introduction: there is the need of more references about CMC and fiber production (see, just for example, Hiroyuki Takeuchi, Kaneo Noake, Tamio Serita's patent). A lot of work has been already done.

*Characterization: number of scans for NMR spectra?

*Results

- Si instead of S, in table 1.
- It is not clear why Si will reduce at high pyrolytic temperatures
- It is declared that from Raman a lot of free C is present, thus it cannot be taken into account in the proposed formula in table 1
- At least in two parts of the paper the porosity of the fiber is mentioned. It should be a good idea to measure N₂ physisorption in order to better define this porosity.
- The amount of the DCP seems very high, so that should be considered as a "reagent". Could the author explain the reason for this choice? Is it detectable with NMR or FTIR? Is it washed away?
- Although the chemical processes are well described and commented, and the final properties of the materials seem valuable, the processes result critical for various reasons: as stated above the amount of the radical initiator, the usefulness of the silazane to produce materials with no or negligible amount of N in the structure, which, moreover, causes the release of NH₃. This by-product, on an industrial scale, will be a problem due to its toxicity.

Reviewer: 3

Comments to the Author(s)

This paper reported Preparation and Structure of SiOCN Fibers Derived from Cyclic Silazane/PAA Hybrid Precursor, which fits the scope of the journal Royal Society Open Science. The paper is straightforward and well-organized. I would recommend publication of this work after minor revision. The introduction of PAA to TTCSZ to create a hybrid precursor remarkably improved spinnability of the silazane and should be widely applicable to other unspinnable PDC preceramic polymers. What happened between the PAA and TTCSZ? Was there a chemical reaction involved? Can the authors show the proof and comment more on this issue?

Author's Response to Decision Letter for (RSOS-190690.R0)

See Appendix A.

RSOS-190690.R1 (Revision)

Review form: Reviewer 2

Is the manuscript scientifically sound in its present form?

No

Are the interpretations and conclusions justified by the results?

Yes

Is the language acceptable?

Yes

Do you have any ethical concerns with this paper?

No

Have you any concerns about statistical analyses in this paper?

No

Recommendation?

Accept with minor revision (please list in comments)

Comments to the Author(s)

The paper in the present form is now improved. The authors have taken into account the suggestions.

The major concern now is the statement about process scalability. I agree with the usefulness of fiber electrospinning but (page 12 lines 47-499 the sentence of "technique easily applicable..." should be modified. Please take into account comments 7 and 8 of reviewer #2. the author should clarify, both in the introduction and conclusions, that the process as it is has some critical points that should be taken in to account for scaling up, but it is however useful for scientists who work with this materials for all the characterization. It can be a starting point somehow. These modifications are mandatory.

Minor point: figure are definitely improved but the vertical dotted lines in the FTIR figures are useless. It can be useful instead put them on the interesting peak in order to drive the readers eyes to the important peaks otherwise discard them.

Decision letter (RSOS-190690.R1)

27-Aug-2019

Dear Mr Ren:

On behalf of the Editors, I am pleased to inform you that your Manuscript RSOS-190690.R1 entitled "Preparation and Structure of SiOCN Fibers Derived from Cyclic Silazane/PAA Hybrid Precursor" has been accepted for publication in Royal Society Open Science subject to minor revision in accordance with the referee suggestions. Please find the referees' comments at the end of this email.

The reviewers and Subject Editor have recommended publication, but also suggest some minor revisions to your manuscript. Therefore, I invite you to respond to the comments and revise your manuscript.

- Ethics statement

- Data accessibility

<http://datadryad.org/submit?journalID=RSOS&manu=RSOS-190690.R1>

- Competing interests

- Authors' contributions

- Acknowledgements

- Funding statement

Because the schedule for publication is very tight, it is a condition of publication that you submit the revised version of your manuscript before 05-Sep-2019. Please note that the revision deadline will expire at 00.00am on this date. If you do not think you will be able to meet this date please let me know immediately.

Kind regards,
Alice Power

Editorial Coordinator
Royal Society Open Science
openscience@royalsociety.org

on behalf of Dr Maria Charalambides (Associate Editor) and R. Kerry Rowe (Subject Editor)
openscience@royalsociety.org

Reviewer comments to Author:
Reviewer: 2

The paper in the present form is now improved. The authors have taken into account the suggestions.

The major concern now is the statement about process scalability. I agree with the usefulness of fiber electrospinning but (page 12 lines 47-499 the sentence of "technique easily applicable..." should be modified. Please take into account comments 7 and 8 of reviewer #2. the author should clarify, both in the introduction and conclusions, that the process as it is has some critical points that should be taken in to account for scaling up, but it is however useful for scientists who work with this materials for all the characterization. It can be a starting point somehow. These modifications are mandatory.

Minor point: figure are definitely improved but the vertical dotted lines in the FTIR figures are useless. It can be useful instead put them on the interesting peak in order to drive the readers eyes to the important peaks otherwise discard them.

Author's Response to Decision Letter for (RSOS-190690.R1)

See Appendix B.

Decision letter (RSOS-190690.R2)

04-Sep-2019

Dear Mr Ren,

I am pleased to inform you that your manuscript entitled "Preparation and Structure of SiOCN Fibers Derived from Cyclic Silazane/PAA Hybrid Precursor" is now accepted for publication in Royal Society Open Science.

on behalf of Dr Maria Charalambides (Associate Editor) and R. Kerry Rowe (Subject Editor)
openscience@royalsociety.org

Appendix A

General Comment Reviewer #1: Based on the aforementioned remarks I recommend this MS for publication.

Response: We appreciate your valuable recommendation.

Reviewer #1 Comment #1: The research is well conducted and the MS is nicely written. However minor topological errors were found, authors are encouraged to correct for example in the introduction (page 2, line 37) “drawing” instead of “drwaing”. In table 1 (page 4) “Si” instead of “S” in the column related to the combustion analysis...

Response: Thank you for your comments. The spelling errors are fixed in the revised manuscript and highlighted in yellow.

CHANGES MADE: “drwaing” to “drawing” in page 2; “S” to “Si” in table 1, page 5.

Reviewer #1 Comment #2: The elemental analysis performed on the hybrid fibers are shown at 2 different pyrolysis temperatures in Table 1. The N content is clearly low and keeps on decreasing with the temperature increase. Even though the authors explained the loss of nitrogen in page 9 paragraph C, which I agree with, the elemental composition presented describes more a SiOC system rather than SiOCN. I suggest that authors include the N content in their final formula if they want to describe their ceramic as SiOCN. I also suggest that they put the NMR signals related to silazanes directly on the spectra to support their argument stated in page 9, line 24: “nonetheless additional minor signals are still present around -17ppm, indicating that silazane bonds are still present”. It will be easier for the reader to keep track.

Response: Thank you for your comments. We will keep the composition as SiOCN in the manuscript. At this moment, we performed NMR spectrum (quick scan) for TTCSZ+DCP system and it is included as inset of Figure 9a in the revised manuscript. From our reference (ref #31 in the revised version), the 6-ring cyclotrisilazane at -14.99 ppm and 8-ring is expected to appear at slightly more negative chemical shift values (around -17 ppm) because of released constraints.

CHANGES MADE: A inset was included in Fig. 9a which shows a quick scan of silazane (TTCSZ) as the reviewer suggested. “not shown” in page 9 is changed to “Fig. 9a inset” and highlighted.

General Comment Reviewer #2: The work is described in details and every step is adequately commented and proved with the above-mentioned techniques.

However, there are some points that must be clarified to give value to the work, especially dealing with the reason of the reagent choice that is a big part.

Response: We appreciate your valuable comments. We have corrected the spelling mistakes and answered the unexplained questions. Please see point by point responses below.

Reviewer #2 Comment #1: Introduction: there is the need of more references about CMC and fiber production (see, just for example, Hiroyuki Takeuchi, Kaneo Noake, Tamio Serita’s patent). A lot of work has been already done.

Response: Thank you for your comments. In the introduction section, we added more contents about production of fibers of PDC material. Highlighted in yellow.

CHANGES MADE: Added new paragraph (shown below) in the introduction (page 2) with production related works cited in the revised manuscript as reference 6-10.

With the rising demand for better processability, higher ceramic yield and enhanced properties of final ceramic product, numbers of new silicon based preceramic polymers were synthesized in the 1980s with modified properties that is sufficient enough for spinning. The large-scale production of ceramic fibers started early as 1990s. Either melt- or dry-spinning was applied to mainly polysilanes or polyborosilazanes to produce non-oxide SiC or SiBCN fibers. In terms of melt-spinning, modified polymer with required viscosity is continuously fed through a heated (about 150 °C) nozzle with hundreds of holes. Fibers are effectively drawn in this way and cured by additional thermal or chemical treatments. Up until recently, there are many types of ceramic fibers developed over 20 years ago and still commercially available from various manufacturers, such as Hi-Nicalon (from COI), Sylramic (from UBE), etc.

Reviewer #2 Comment #2: Characterization: number of scans for NMR spectra?

Response: Thank you for your comments. The number of scans is included in all the NMR related figure captions, which are highlighted in Fig. 5, Fig. 6 and Fig. 9.

CHANGES MADE: Number of scans included in Fig. 5 (page 7), Fig. 6 (page 7) and Fig. 9 (page 10) captions.

Reviewer #2 Comment #3: Si instead of S, in table 1.

Response: Thank you for your comments. The spelling errors are fixed in the revised manuscript and highlighted in yellow.

CHANGES MADE: “S” to “Si” in table 1, page 5.

Reviewer #2 Comment #4: It is not clear why Si will reduce at high pyrolytic temperatures

Response: Thank you for pointing this out. It turned out to be that carbon content in 700 was a machine error from the combustion analysis. Thus, we removed the table 1 content and replaced with XPS depth profile (15 min via Ar) survey information. The Si, C, O and N contents are shown in atomic percentage in the revised table 1.

Pyrolysis Temperature (°C)	Element atomic percentage (at. %)			
	C	Si	O	N
700	71.2	8.6	19.4	0.3
800	50.8	15.5	33.4	0.4

CHANGES MADE: Table 1 contents are replaced by XPS depth profile information in the manuscript (page 5). Table 1 caption changed from “Combustion analysis of samples in weight percentage” to “Chemical composition of fibers by XPS depth-profiling survey scan in atomic percentage.”

Corresponding combustion analysis information removed from Characterization section. Corresponding sentence “Undesired significant silicon loss occurs at 800 °C that leads to a great increase in carbon and oxygen percentage” is removed (page 4). Sentence “Because Si-C bonds are sp^3 , stronger D1, D4-bands in 700 sample confirms higher Si contents previously reported by the combustion analysis” (page 6) modified to “Because Si-C bonds are sp^3 , stronger D1, D4-bands in 700 sample confirms higher SiO_2C_2 and SiO_3C composition discussed below with the NMR analysis (Fig. 5a).” and highlighted

Reviewer #2 Comment #5: It is declared that from Raman a lot of free C is present, thus it cannot be taken into account in the proposed formula in table 1

Response: Thank you for your comments. We agree that this formula cannot fully represent the structure in our fibers. Therefore, we remove the entire column from the table in the revised manuscript.

CHANGES MADE: The entire “Formula” column is removed from the Table 1 (page 5). Corresponding text “with formula obtained by neglecting hydrogen and nitrogen contents after pyrolysis” (page 4) also removed.

Reviewer #2 Comment #6: At least in two part of the paper the porosity of the fiber is mentioned. It should be a good idea to measure N_2 physisorption in order to better define this porosity.

Response: Thank you for your comments. We have mentioned porosity twice in the manuscript in order to describe the surface feature of the fibers from SEM images. The surface feature is obtained only from the microscopic observation while we wanted to show the fiber diameter uniformity. We did not perform further BET analysis because this surface characteristic was not our main concern for the fiber application. In addition, we think the fibers are slightly porous on the surface and solid inside. Therefore, we consider that BET analysis may not be as helpful as fully porous materials.

CHANGES MADE: None.

Reviewer #2 Comment #7: The amount of the DCP seems very high, so that should be considered as a “reagent”. Could the author explain the reason for this choice? Is it detectable with NMR or FTIR? Is it washed away?

Response: Thank you for your comments.

1. It is true that we used higher amount (10 wt. %) of DCP in our initial TTCSZ solution when compared to other published works (usually ~1 wt. %) from our lab[1] and elsewhere[2-4]. The reason for this was hoping to improve the crosslinking behavior of the TTCSZ/PAA hybrid system. We introduced PAA as spinning agent, such that TTCSZ molecules are likely to be embedded in a PAA environment. Also, we know that crosslinking of TTCSZ will reduce the mobility of the TTCSZ molecules (NMR evidence). Considering all above, we decided to apply high amount of DCP to ensure as much of TTCSZ molecules get crosslinked with each other.
2. & 3. We performed additional FTIR analysis to pure crosslinker dicumyl peroxide (DCP), TTCSZ + DCP (10 wt. %) mixture and crosslinked TTCSZ + DCP (10 %). The result shows that DCP is not obviously detectable from the FTIR together with TTCSZ (the signature peaks are extremely

weak). Especially after crosslinking at 160 C, DCP signal became completely undetectable by FTIR. Hence, fair to assume that DCP is washed away or decomposed at crosslinking stage.

CHANGES MADE: The above characterization and related discussion is included as supplemental material. Corresponding sentence “The weak influence of DCP on FT-IR signal is discussed in supplemental material (Fig. S1)” added in the revised manuscript (page 8).

Mistakes in labeling FTIR peaks in Fig.8 (page 9) is corrected in the revised manuscript.

Reviewer #2 Comment #8: Although the chemical processes are well described and commented, and the final properties of the materials seems valuable, the processes result critical for various reason: as stated above the amount of the radical initiator, the usefulness of the silazane to produce materials with no or negligible amount of N in the structure, which, moreover, causes the release of NH₃. This by-product, on an industrial scale, will be a problem due to its toxicity.

Response: Thank you for your comments. We also did not expect this circumstance would happen when we firstly started the series of experiments. What we did, instead of simply switching our preceramic polymer to siloxane, was performing additional experiments such as NMR and FTIR at each stage to dug deeper into the loss of nitrogen. As we mentioned in the introduction part, the properties of this cyclic silazane is remain unrevealed compare to the siloxane with similar structure, hence, we consider our work is also a good investigation of this type of silazane that is worthy published as well.

CHANGES MADE: None.

General Comment Reviewer #3: This paper reported Preparation and Structure of SiOCN Fibers Derived from Cyclic Silazane/PAA Hybrid Precursor, which fits the scope of the journal Royal Society Open Science. The paper is straightforward and well-organized. I would recommend publication of this work after minor revision.

Response: We appreciate your valuable recommendation.

Reviewer #3 Comment #1: What happened between the PAA and TTCSZ? Was there chemical reaction involved? Can the authors show the proof and comment more on this issue?

Response: Thank you for your comments. We have stated in the manuscript that PAA showed very weak crosslinking activity. The statement was supported by NMR (Fig. 9). Fig. 9b ^{13}C CP MAS NMR suggests no significant changes in C=O and -CH(COOH)-CH₂ signals, which are the main functional groups of PAA.

We performed FTIR of pure TTCSZ (10 wt% DCP) and crosslinked pure TTCSZ (10 wt% DCP). The result is already shown once above (**Reviewer #2 Comment #7**). When comparing this result with previous FTIR results, several points can be made here:

- (1) Neat raw PAA and heated PAA at 160 °C does not show significant changes from FTIR spectrum.
- (2) Comparing TTCSZ/PAA crosslinked at 160 °C and TTCSZ crosslinked at 160 °C without PAA, the main differences are where PAA peaks at (C=O, CH₂ and C-O peak, circled in the figure).

Based on the abovementioned points. We conclude that introduction of PAA did not alter or hardly alter very little to TTCSZ FTIR signal. Hence, we think there was no chemical reaction took place between PAA and TTCSZ. We will include this statement in the manuscript and include all investigation in the supplemental material.

CHANGES MADE: The above characterization and related discussion is included as supplemental material. Corresponding sentence “Further discussion regarding crosslinking behavior between PAA and TTCSZ is presented in supplemental material (Fig. S2)” added in the revised manuscript (page 10).

Reference

- 1 David, L., Bhandavat, R., Barrera, U., Singh, G. 2016 Silicon oxycarbide glass-graphene composite paper electrode for long-cycle lithium-ion batteries. *Nature Communications*. **7**, (10.1038/ncomms10998)
- 2 Ahn, D., Raj, R. 2011 Cyclic stability and C-rate performance of amorphous silicon and carbon based anodes for electrochemical storage of lithium. *Journal of Power Sources*. **196**, 2179-2186. (10.1016/j.jpowsour.2010.09.086)
- 3 Jana, R. N., Mukunda, P. G., Nando, G. B. 2003 Thermogravimetric analysis of compatibilized blends of low density polyethylene and poly(dimethyl siloxane) rubber. *Polymer Degradation and Stability*. **80**, 75-82. (10.1016/s0141-3910(02)00385-3)
- 4 Schiavon, M. A., Soraru, G. D., Yoshida, I. V. P. 2002 Synthesis of a polycyclic silazane network and its evolution to silicon carbonitride glass. *Journal of Non-Crystalline Solids*. **304**, 76-83. (10.1016/s0022-3093(02)01007-4)

Appendix B

Reviewer #2 Comment #1:

the paper in the present form is now improved. The authors have taken into account the suggestions. the major concern now is the statement about process scalability. I agree with the usefulness of fiber electrospinning but (page 12 lines 47-499 the sentence of "technique easily applicable..." should be modified. Please take into account comments 7 and 8 of reviewer #2. the author should clarify, both in the introduction and conclusions, that the process as it is has some critical points that should be taken in to account for scaling up, but it is however useful for scientists who work with this materials for all the characterization. It can be a starting point somehow. These modifications are mandatory.

Author Response: Thank you for your valuable opinion. The sentence is modified to discuss the value point within research rather than scalability.

CHANGES MADE:

1. New sentence added in the introduction:

“The hybrid approach involving hand spinning process is highly efficient for lab scale testing and to develop fundamental understanding related to processing of preceramic fibers and related polymer to ceramic transformations without the need for complex fiber drawing equipment.”

2. The old sentence “This hybrid technique can be easily applicable to industrial scale manufacturing and not be limited to this specific type of silazane (should be equally applicable to other silazanes and siloxanes). The fibers and technique are highly recommended to high cost-effective CMC materials.”

has been modified to

“This hybrid technique is recommended to lab scale early production of new types of ceramic fibers. High productivity of this technique can easily satisfy various characterization needs with minimum setup requirement.” The fiber products are suggested as reinforcement for CMC materials.”

Reviewer #2 Comment #2:

Minor point: figure are definitely improved but the vertical dotted lines in the FTIR figures are useless. It can be useful instead put them on the interesting peak in order to drive the readers eyes to the important peaks otherwise discard them.

Response: Thank you for your comment. The broken line was initially introduced for the readers to reduce the difficulty of roughly identifying peak location.

CHANGES MADE:

The dotted lines have been removed from FTIR plot (Figure. 8, Figure. S1, and S2) in the revised manuscript.